# Latent Intrinsics Emerge from Training to Relight

Xiao Zhang[1]    William Gao[1]    Seemandhar Jain[2]    Michael Maire[1]
D.A. Forsyth[2]    Anand Bhattad[3]

[1]University of Chicago    [2] University of Illinois Urbana Champaign
[3]Toyota Technological Institute at Chicago

zhang7@uchicago.edu    bhattad@ttic.edu

https://latent-intrinsics.github.io/

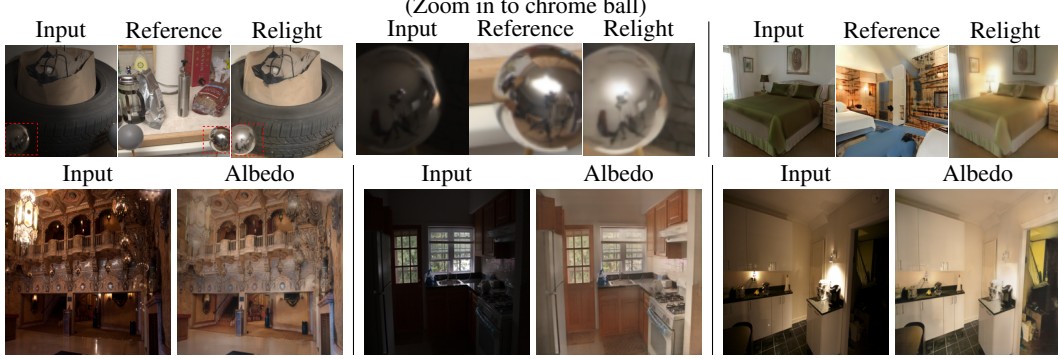

Figure 1: We describe a purely data-driven image relighting model. Our model recovers latent variables representing scene intrinsic properties from one image, latent variables representing lighting from another, then applies the lighting to the intrinsics to produce a relighted scene (**top row**). There is no physical model of intrinsics, extrinsics or their interaction. Our model relights images of real scenes with SOTA accuracy and is more accurate than current supervised methods. Note how, for the chrome ball detail in **top center**, the specular reflections on the chrome ball (which give an approximate environment map) change when the extrinsics are changed. Note how our model ascribes lighting to visible luminaires when it can (**top right**), despite the absence of any physical model. A physical model accounts only for effects in that model, and most physical models of surfaces are approximate; in contrast, a latent intrinsic model accounts for whatever produces substantial effects in training data. Latent intrinsics yield albedo in a natural fashion (light the scene with an appropriate illuminant). **Bottom row** shows SOTA albedo estimates recovered from our latent intrinsics.

## Abstract

Image relighting is the task of showing what a scene from a source image would look like if illuminated differently. Inverse graphics schemes recover an explicit representation of geometry and a set of chosen intrinsics, then relight with some form of renderer. However error control for inverse graphics is difficult, and inverse graphics methods can represent only the effects of the chosen intrinsics. This paper describes a relighting method that is entirely data-driven, where intrinsics and lighting are each represented as latent variables. Our approach produces SOTA relightings of real scenes, as measured by standard metrics. We show that albedo can be recovered from our latent intrinsics without using any example albedos, and that the albedos recovered are competitive with SOTA methods.

38th Conference on Neural Information Processing Systems (NeurIPS 2024).

# 1 Introduction

Relighting – taking an image of a scene, then adjusting it so it looks as though it had been taken under another light – has a range of applications, including commercial art (e.g., photo enhancement) and data augmentation (e.g., making vision models robust to varying illumination). As a technical problem, relighting is very hard indeed, likely because how a scene changes in appearance when the light is changed can depend on complex surface details (grooves in screws; bark on trees; wood grain) that are hard to capture either in geometric or surface models.

One common approach to relighting a scene is to infer scene characteristics (geometry, surface properties) using inverse graphics methods, then render the scene with a new light source. This approach is fraught with difficulties, including the challenge of selecting which material properties to infer and managing error propagation. These methods perform best in outdoor scenes with significant shadow movements but struggle with indoor scenes where interreflections create complex effects (Section 4.2).

As this paper demonstrates, a purely data-driven method offers an attractive alternative. A source scene, represented by an image, is encoded to produce a latent representation of intrinsic scene properties. A source illumination, represented by another image, is encoded to produce a latent representation of illumination properties. These intrinsic and extrinsic properties are combined and then decoded to produce the relighted image. As a byproduct of this training, we find that the latent representation of intrinsic scene properties behaves like an albedo, while another latent representation acts as a lighting controller.

Our model can capture complex scene characteristics without explicit supervision by capturing intrinsic properties as latent phenomena, making it particularly appealing. In contrast to a physical model, we are not required to choose which effects to capture. This latent approach reduces the need for detailed geometric and surface models, simplifies the learning process, and enhances the model's ability to generalize to diverse and unseen scenes. This makes it highly applicable to a wide range of real-world scenarios.

**Contributions:** We present the first fully data-driven relighting method applicable to images of real complex scenes. Our approach requires no explicit lighting supervision, learning to relight using paired images alone. We demonstrate that this method effectively trains and generalizes, producing highly accurate relightings. Furthermore, we demonstrate that albedo-like maps can be generated from the model without supervision or prior knowledge of albedo-like images. These intrinsic properties emerge naturally within the model. We validate our model on a held-out dataset, applying target lighting conditions from various scenes to assess its generalization capability and precision in real-world scenarios (Section 4.2).

# 2 Related Work

**Intrinsic Images.** Humans have been known to perceive scene properties independent of lighting since at least 1867 [46, 21, 4, 20]. In computer vision, the idea dates to Barrow and Tenenbaum [3] and comprises at least depth, normal, albedo, and surface material maps. Depth and normal estimation are now well established (eg [24]). There is a rich literature on albedo estimation (dating to 1959 [30, 31]!). A detailed review appears in [16], which breaks out methods as to what kinds of training data they see. Early methods do not see any form of training data, but more recently both CGI data and manual annotations of relative lightness (labels) have become available. Early efforts, such as SIRFS [1], focused on using shading information to recover shape, illumination, and reflectance, highlighting the importance of modeling these factors for intrinsic image analysis. Recent strategies include: deep networks trained on synthetic data [33, 23, 15]; and conditional generative models [29].

The weighted human disagreement ratio (WHDR) evaluation framework was introduced by [5] using the IIW dataset. This is a dataset of human judgments that compare the absolute lightness at pairs of points in real images. Each pair is labeled with one of three cases (first lighter; second lighter; indistinguishable) and a weight, which captures the certainty of labelers. One evaluates by computing a weighted comparison of algorithm predictions with human predictions; WHDR scores can be improved by postprocessing because most methods produce albedo fields with very slow gradients, rather than piecewise constant albedos. [10] demonstrate the value of "flattening" albedo (see also [39]); [11] employ a fast bilateral filter [2] to obtain significant improvements in WHDR.

**Using Intrinsic Images for Relighting.** Bhattad and Forsyth [6] demonstrated that intrinsic images could be used for reshading inserted objects. This approach can be extended by adjusting the shading in both the foreground and background to eliminate discrepancies [12]. Intrinsic images and geometry-aware networks have been used for multi-view relighting [41]. StyLitGAN [9] introduced a method to relight images by identifying directional vectors in the latent space of StyleGAN, but can only relight StyleGAN generated images and requires explicit albedo and shading to guide relighting. It can be extended to real images using a GAN inversion, but does not generalize [7]. LightIt [28] controls lighting changes in image generation using diffusion models, by conditioning on shading and normal maps to achieve consistent and controllable lighting. Like these methods, we use intrinsics and extrinsics to relight, but ours are latent, with no explicit physical meaning.

**Color Constancy.** Image color is ambiguous: a green pixel could be the result of a white light on a green surface, or a green light on a white surface. Humans are unaffected by this ambiguity (eg [21, 4]; recent review in [50]). There is extensive computer vision literature; a recent review appears in [32]. We do not estimate illumination color but estimate a single color correction (Section 4.2).

**Lighting Estimation and Representation.** Accurate lighting representation is crucial for tasks like object insertion and relighting. Traditional methods used parametric models such as environment maps and spherical harmonics to represent illumination [13, 42]. Debevec's seminal work [13] on recovering environment maps from images of mirrored spheres set the foundation for many subsequent works. Methods by Karsch et al. [26, 27], Gardner et al. [17, 18], Garon et al. [19] and Weber at al. [49] advanced the field by using learned models to recover parametric, semi-parametric or panoramic representations of illumination. Recent approaches include representing illumination fields as dense 2D grids of spherical harmonic sources [34, 36] or learning 3D volumes of spherical Gaussians [48]. These methods can model complex light-dependent effects but require extensive CGI datasets for training [43, 35]. Our approach diverges by not relying on labeled illumination representations or CGI data, instead producing abstract representations of illumination through deep features without specific physical interpretations.

**Image-based Relighting.** Other works focus on portrait relighting using deep learning [45, 55, 40, 44], which are typically specialized to faces and trained on paired or light-stage data. Self-supervised methods for outdoor image relighting leverage single-image decomposition with parametric outdoor illumination, benefiting from simpler lighting conditions dominated by sky and sunlight [54, 37]. [22] introduced a self-attention autoencoder model to re-render a source image to match the illumination of a guide image, focusing on separating scene representation and lighting estimation with a self-attention mechanism for targeted relighting. Similarly, [51] proposed a depth-guided image relighting, which combines source and guide images along with their depth maps to generate relit images. In contrast, our work shows that intrinsic properties relevant to relighting can emerge naturally from training to relight, facilitating complex scene relighting without the need for explicit lighting estimation. We compare with both [22] and [51] for relighting capabilities on real scenes.

**Emergent Intrinsic Properties.** Bhattad et al. [8] and Du et al. [14] demonstrate that intrinsic images can be extracted from generative models using a small intrinsic image dataset obtained from pretrained off-the-shelf intrinsic image models. Our work explores how intrinsic image properties emerge as a result of training a model for relighting, without the need for an intrinsic image dataset.

## 3  Learning Latent Intrinsic from Relighting.

Our relighting model can be seen as a form of autoencoder. One encoder computes a latent representation of scene intrinsics from an image of a target scene; another computes a latent representation of scene extrinsics from an image of a placeholder scene in the reference lighting. These are combined, then decoded into a final image of the target scene in the reference lighting. Losses impose the requirements that (a) the final image is right and (b) the latent intrinsics computed for a scene are not affected by illumination. The procedure for combining intrinsics and extrinsics is carefully designed to make it very difficult for intrinsic features of the placeholder scene to "leak" into the final image.

### 3.1  Model structure

**Decoder setup:** Write $\boldsymbol{I}_s^l \in \mathbf{R}^{H \times W \times 3}$ for the input image, captured from scene $s$ with lighting configuration $l$. Training uses pairs $\boldsymbol{I}_s^{l_1}$ and $\boldsymbol{I}_s^{l_2}$, representing the same scene $s$ under different lighting

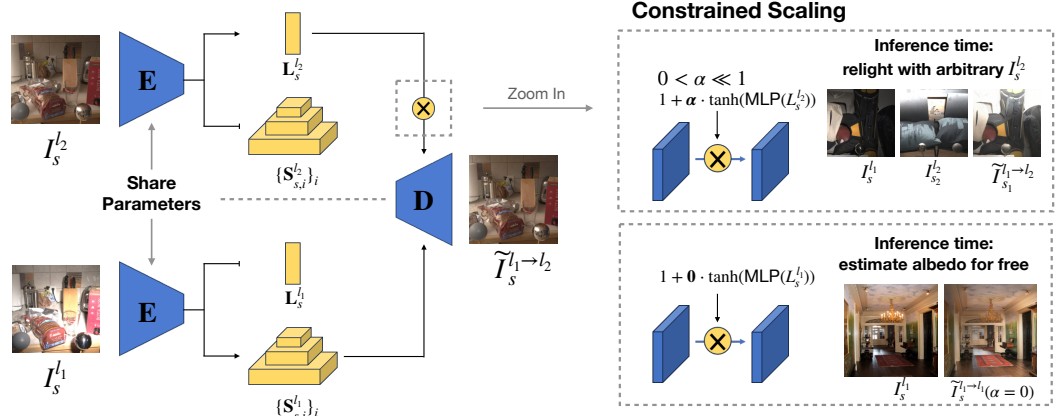

Figure 2: The network diagram of our relighting model. The model functions as an autoencoder, comprising an encoder $E$ and a decoder $D$. **Left Half**: The encoder $E$ maps input image $I_s^l$, captured under scene $s$ and lighting $l$, to low-dimensional extrinsic features $L_s^l$ and set of intrinsic features map $\{S_{s,i}^l\}_i$. The decoder $D$ then generates new images based on these intrinsic and extrinsic representations. **Right Half**: We employ *constrained scaling* for the injection of $L_s^l$, utilizing $0 < \alpha \ll 1$ to regularize the information passed from $L_s^l$, thereby enforcing a low-dimensional parameterization of the extrinsic features. We train our system to relight target images given input paired with images captured under the same scene $s$. During inference, our model demonstrates the ability to generalize to arbitrary reference images for relighting and can estimate albedo for free.

conditions $l_1$ and $l_2$. The model *does not see* detailed lighting information (for example, the index of the lighting) during training, because standardizing lighting settings across various scenes is often impractical.

Write $E$ for the encoder, $D$ for the decoder. The encoder must produce the intrinsic and extrinsic representations from the input image. Write $S_{s,i}^l \in \mathbf{R}^{(H_i \times W_i) \times C_i}$ for spatial feature maps yielding the intrinsic representation, with $i$ for the layer index, and $L_s^l \in \mathbf{R}^C$ for extrinsic features; we have:

$$E(I_s^l) := \{S_{s,i}^l\}_i, L_s^l \tag{1}$$

We apply L2 normalization along the feature channel to both sets of features. During training, we add random Gaussian noise to the input image to enhance semantic scene understanding capabilities:

$$E(I_s^l + \sigma\epsilon) := \{S_{s,i}^l\}_i, L_s^l \tag{2}$$

**Decoder setup:** The decoder $D$ relights $I_s^{l_1}$ using extrinsic features extracted from $I_s^{l_2}$:

$$D(\{S_s^{l_1}\}, L_s^{l_2}) := \widetilde{I}_s^{l_1 \to l_2} \tag{3}$$

We optimize the autoencoder using a pixel-wise loss on both relighted and reconstructed images:

$$\mathcal{L}_{\text{relight}} := \mathcal{L}_{\text{pixel}}(\widetilde{I}_s^{l_1 \to l_2}, I_s^{l_2}) + \mathcal{L}_{\text{pixel}}(\widetilde{I}_s^{l_2 \to l_2}, I_s^{l_2}) \tag{4}$$

where $\mathcal{L}_{\text{pixel}}$ represents the pixel-wise losses: L2 distance on pixels; structural similarity index (SSIM) [47]; and l2 distance on image spatial gradient (weights 10, 0.1 and 1 respectively).

## 3.2 Intrinsicness

**Intrinsicness:** Our model should report the same latent intrinsic for the same scene in different lightings, so we apply the following loss to the encoder:

$$\mathcal{L}_{\text{intrinsic}} := \sum_i \|S_{s,i}^{l_1} - S_{s,i}^{l_2}\|_2 + 1\text{e-}3 \cdot \mathcal{L}_{\text{reg}}(S_{s,i}^{l_1}) \tag{5}$$

where $\mathcal{L}_{\text{reg}}$ is a regularization term on intrinsic features, defined as follows:

$$\mathcal{L}_{\text{reg}}(S) := \|R(S) - R(\widehat{S})\|_2 \tag{6}$$

$$R(S) := \log\det\left(I + \frac{d}{n\lambda^2}S^\top S\right) \tag{7}$$

where $R(\boldsymbol{S})$ is the coding rate [53] for a matrix $\boldsymbol{S} \in \mathbb{R}^{n \times d}$ with each row l2 normalized, under a distortion constant $\lambda$. $\widehat{\boldsymbol{S}}$ is a random matrix with the same shape of $\boldsymbol{S}$ and each row of $\widehat{\boldsymbol{S}}$ is sampled from uniform hyperspherical distribution at the start of learning. In Eqn.5, $R(\widehat{\boldsymbol{S}})$ serves as the optimization target of $R(\boldsymbol{S})$ to encourage the $\boldsymbol{S}$ to uniformly spread out in the hyperspherical space. This strategy is now widely used in self-supervised learning; without the regularization term, the model can minimize the feature distance by simply collapsing the distribution of $\boldsymbol{S}_{s,i}^l$ with small variance, which will not yield effective lighting invariance.

## 3.3 Combining intrinsics and extrinsics

The placeholder scene is necessary to communicate illumination to the model, but has important nuisance features. Intrinsic information from this scene could "leak" into the final image, spoiling results. We introduce *constrained scaling*, a structural bottleneck that restricts the amount of information transmitted from the learned extrinsic features.

Write $\boldsymbol{F} \in \mathbf{R}^{h \times w \times c}$ for the feature map fed to the decoder. Constrained scaling combines intrinsic and extrinsic features by

$$\widetilde{\boldsymbol{F}} := \boldsymbol{F} \odot \left(1 + \alpha \cdot \tanh\left(\mathrm{MLP}\left(\boldsymbol{L}_s^l\right)\right)\right) \tag{8}$$

where MLP, a series of fully connected layers with non-linear activation, aligns $\boldsymbol{L}_s^l$ to the latent channel dimension of $\boldsymbol{F}$ and $\alpha \ll 1$ is a small non-negative scalar (we use 5e-3). This approach means that any single extrinsic feature vector has little effect on the feature – for an effect, the extrinsics must be pooled over multiple locations. Illumination fields tend to be spatially smooth, supporting the insight that enforced pooling is a good idea.

Constrained scaling compresses latent vectors into a very small numerical range, making learning difficult. We use a regularizer to promote a uniform distribution of $\boldsymbol{L}_s^l$, which improves optimization. In particular, we have

$$\mathcal{L}_{\mathrm{extrinsic}} := \mathcal{L}_{\mathrm{reg}}(\boldsymbol{L}_s^l) \tag{9}$$

By choosing $\alpha \ll 1$ and training model with uniform regularization term Eqn.9, we effectively push the lighting code to uniformly spread over $[-\alpha, \alpha]$ where the absolute value of each channel indicates the strength of the light. As a side effect, by setting $\alpha = 0$ to disable the contribution of the lighting code, we get image albedo estimation from our model for free.

Our final training objective is weighted combination of all individual loss terms:

$$\mathcal{L} := \mathcal{L}_{\mathrm{relight}} + 1\mathrm{e}{-1} \cdot \mathcal{L}_{\mathrm{intrinsic}} + 1\mathrm{e}{-4} \cdot \mathcal{L}_{\mathrm{extrinsic}} \tag{10}$$

# 4 Experiments

We will first provide a brief description of our experimental procedure (Sec 4.1), followed by a discussion on how we evaluate the various relighting capabilities of our approach, including its strong generalization across datasets with different distributions (Sec 4.2). Finally, we will present the emergent albedo that is recovered from the latent intrinsic without using any albedo-like images (Sec 4.3).

## 4.1 Experiment Details

**Training Details** We train our model using the MIT multi-illumination dataset [38], which includes images of 1,015 indoor scenes captured under 25 fixed lighting, totaling 25,375 images. We follow the official data split and train our model on the 985 training scenes. During training, we randomly sample pairs of images from the same scene under different lighting conditions and perform random spatial cropping, with the crop ratio randomly selected between 0.2 and 1.0, followed by resizing the cropped image to a resolution of 256x256. For further details, please refer to our appendix.

## 4.2 Evaluating image relighting

**Relighting on the Multi-illumination dataset:** We relight images of scenes in the test set using reference images from the test set, then compare to the correct known relighting from the test set

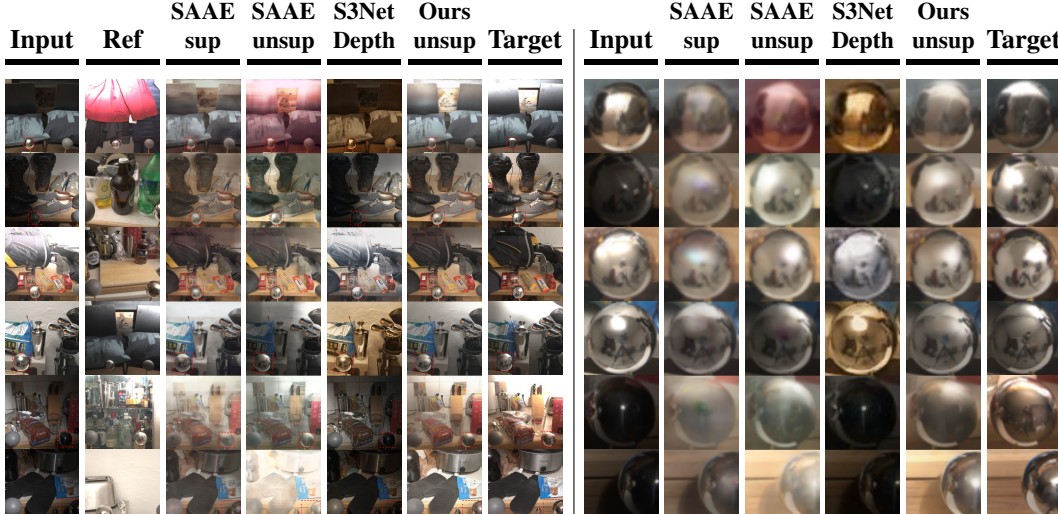

|     | SAAE sup | SAAE unsup | S3Net Depth | Ours unsup | Target |     | SAAE sup | SAAE unsup | S3Net Depth | Ours unsup | Target |

Figure 3: Our method outperforms all other approaches in estimating light and rendering the scene. The Unsupervised SA-AE [22] method fails by incorporating intrinsic elements from reference images. The S3Net [51] approach struggles with rendering when using unpaired reference images. ***Right***: A zoomed-in view of the chrome ball was used as a probe to evaluate detail preservation in the environment map. Our method effectively retains the intricate room layout and accurately renders the appropriate lighting effects.

| Methods | Labels | Raw Output | | Color Correction | |
|---|---|---|---|---|---|
| | | RMSE↓ | SSIM↑ | RMSE↓ | SSIM↑ |
| Input Img | - | 0.384 | 0.438 | 0.312 | 0.492 |
| SA-AE [22] | Light | **0.288** | **0.484** | 0.232 | 0.559 |
| SA-AE [22] | - | 0.443 | 0.300 | 0.317 | 0.431 |
| S3Net [51] | Depth | 0.512 | 0.331 | 0.418 | 0.374 |
| S3Net [51] | - | 0.499 | 0.336 | 0.414 | 0.377 |
| Ours($\sigma = 0$) | - | 0.326 | 0.232 | 0.242 | 0.541 |
| Ours(w/o $\mathcal{L}_{reg}$) | - | 0.315 | 0.462 | 0.232 | 0.550 |
| Ours | - | 0.297 | 0.473 | **0.222** | **0.571** |

Table 1: We assess the quality of image relighting using the multi-illumination dataset [38]. Our method, when evaluated on raw output, significantly outperforms all other unsupervised approaches and achieves competitive results compared to the supervised SA-SA [22], which requires ground truth light supervision. When we correct the colors by eliminating global color drift caused by light ambiguity, our method surpasses all other approaches. Additionally, warming up the model as a denoising autoencoder proves beneficial compared to when it is not warmed up ($\sigma = 0$).

| $\alpha$ | Raw Output | | Color Correction | |
|---|---|---|---|---|
| | RMSE ↓ | SSIM ↑ | RMSE↓ | SSIM↑ |
| $\infty$ | 0.471 | 0.287 | 0.352 | 0.407 |
| 1e-2 | 0.314 | 0.444 | 0.238 | 0.546 |
| 5e-3 | **0.297** | **0.473** | **0.222** | **0.571** |
| 1e-3 | 0.312 | 0.453 | 0.256 | 0.524 |
| 5e-4 | 0.309 | 0.460 | 0.253 | 0.533 |

Table 2: We analyze the impact of $\alpha$ on relighting quality using the multi-illumination dataset [38]. Setting $\alpha$ to $\infty$, which removes the scaling constraints, results in poor relighting quality, indicating that restricting information from extrinsic sources significantly improves generation quality. Within a limited parameter search, 5e-3 yields the best results.

using various metrics. For each input image, we randomly sample reference images from different scenes and lighting conditions. To reduce the effect of randomness in comparing different relighting strategies, we select 12 random reference images for each input image, and maintain the same image-reference pairs when evaluating different models. We report the results, measured in RMSE and SSIM, in Table 1. We report these metrics both for absolute predictions and for predictions where any global color shift is corrected by a single, least-squares scale of each predicted color layer (i.e. one scale for R; one for G; one for B). This color correction allows us to distinguish between spatial errors and global color shifts; these appear to have a significant effect, possibly because there are visible color shifts present in some of the dataset images.

In Table 1, we compare to SA-AE [22], a model that requires a ground truth light index for supervision, and S3-Net, which needs a ground truth depth map as a conditional input. For S3-Net, we use a state-of-the-art depth estimator to provide pseudo-GT on the relighting dataset as input. For a fair comparison to our model, which does not require any supervision outside of the ground truth

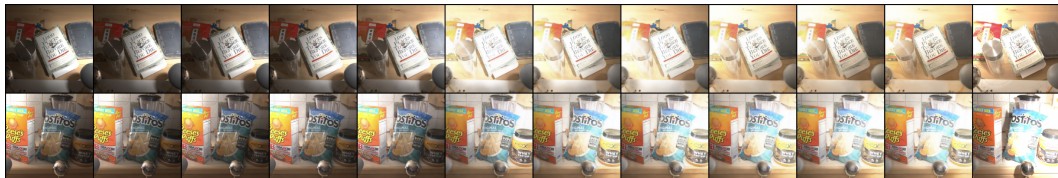

Figure 4: Latent extrinsics can be interpolated successfully; **leftmost** and **rightmost** columns are images from the multi-illumination dataset, and intermediate images are obtained by linear interpolation on the latent extrinsics (light-dependent representations), then decoding. Note how the light seems to "move" across space.

relighting, we also report results for modified versions of the baselines trained without additional supervision. For SA-AE, we train their light estimation model and relighting model end-to-end by removing the loss from light supervision. For S3-Net [51], we simply remove the depth from the model's input.

Without color correction, only light-supervised SA-AE slightly outperforms our model, while all other baselines are significantly worse. The unsupervised version of SA-AE performs much worse because their light estimator struggles to distinguish the extrinsic from the intrinsic components. Specifically, SA-AE also parameterizes the extrinsic as a lower-dimensional representation but without the constrained scaling that our model uses. As a result, the estimated extrinsic from their unsupervised model also carries intrinsic information, and one can see "leaks". S3-Net performs worse in both versions since they concatenate input and reference images before feeding them into the models, which significantly affects the model's generalization ability, especially during test time when we use images from different scenes as references.

On color-corrected images, our approach outperforms all methods, including the light-supervised version of SA-AE, indicating that, up to the constant color drift, our extrinsic estimation network is at least as good as, or even better than, a light estimation network trained with supervision. Removing the denoising setup from our model ($\sigma = 0$) results in worse performance in both cases due to inferior semantic scene understanding. We additionally provide ablation studies on the choices of $\alpha$ in Table 2 and find $\alpha = 5e - 3$ produces the best results.

Each image in the multi-illumination dataset shows a chrome ball, which gives a good estimate of an environment map for that image. Correctly rendering the effects of lighting changes on these chrome balls appears to be extremely difficult; the changes are substantial, and concentrated in a small region of the image (so correct representation of these changes has little effect on typical image losses). Figure 3 shows a crop of our results around this chrome ball. Our method represents these changes well; we are aware of no other results reported for this effect. Compared to other approaches, our model accurately preserves the room layout, even in cases of extreme light changes.

Unlike classical rendering models that use a specific parameterized form to represent extrinsics, our framework learns an implicit extrinsic representation. However, we can still parameterize the learned extrinsic representation to create new light sources. In Figure 4, we demonstrate this capability by rendering images using interpolated extrinsic representations.

**Relighting synthetically relighted images from StyLitGAN:** StyLitGAN [9] is a recent method that can produce multiple illuminations of a single generated room scene by manipulating StyleGAN latents appropriately. In the multi-illumination dataset, reference light and target images tend to share a strong spatial correlation in light patterns. In contrast, StyLitGAN generates extremely challenging images where very significant changes in lighting occur. Furthermore, StyLitGAN images have visible luminaires. To relight the input, the model must infer high-level concepts rather than simply copying the spatially corresponding light patterns from the reference. We train our model using StyLitGAN images to evaluate generalization qualitatively (quantitative evaluation would be of dubious value, because StyLitGAN images are generated rather than real). Figure 5 shows results. Notice how our method successfully relights from references, achieves brighter illuminations by turning on luminaires (here bedside lights), achieves darker scenes by turning off luminaires, and is somewhat less inclined to invent luminaires than StyLitGAN is. The model knows that light must come from somewhere, and how the effects of light are distributed.

**Zero-Shot Relighting:** In Figure.6, we show our model's strong generalization by applying the model solely trained on multi-illumination dataset—without additional training or fine-tuning—to

| StyleGAN Generation | Ref | Ours | StyLitGAN Relight | StyleGAN Generation | Ref | Ours | StyLitGAN Relight |

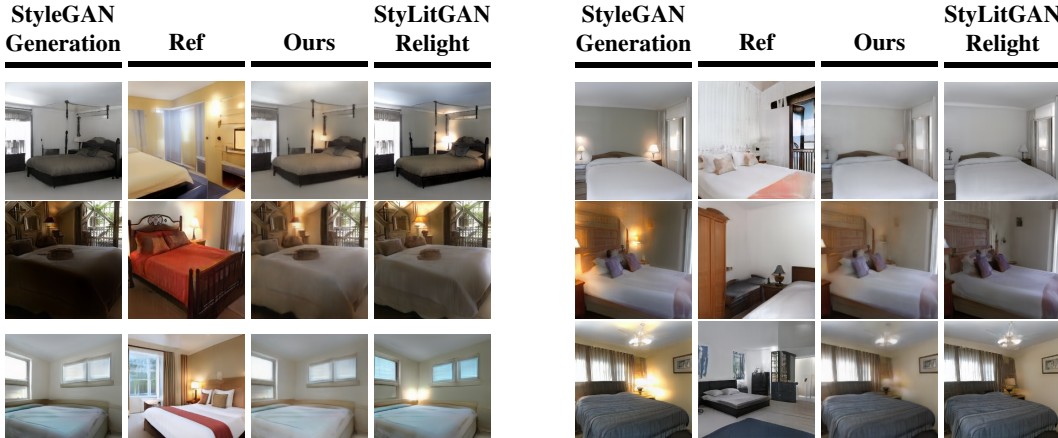

Figure 5: Qualitative results for relighting interior scenes using our relighter trained on images obtained from StyLitGAN (which produces multiple illuminations of a generated scene). StyLitGAN has a strong tendency to increase or decrease illumination by adjusting luminaires, typically bedside lights but also light coming through French windows, etc. On the **left**, where the reference lighting tends to be brighter and more concentrated, notice how for the two top images, our relighter has identified and "turned up" the bedside lights; for the third, it has resisted StyLitGAN's tendency to invent helpful luminaires (there isn't a bedside light where StyLitGAN imputed one, as close inspection shows). On the **right**, where the reference lighting is much more uniform, our relighter has achieved this by "turning down" bedside lights. This is an emergent phenomenon; the method is not supplied with any explicit luminaire model or labeled data.

| Input | Ref | Relight | Input | Ref | Relight | Input | Ref | Relight |

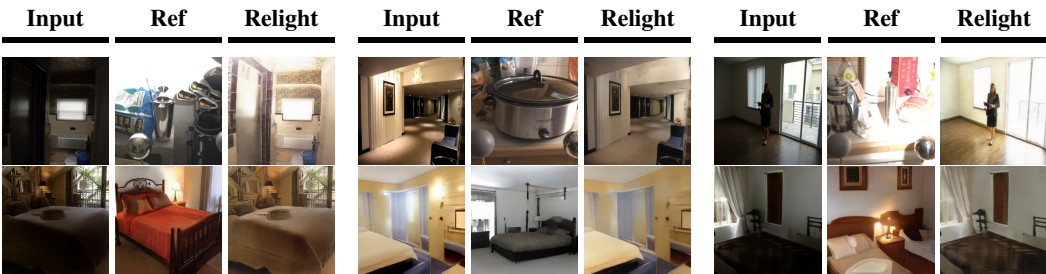

Figure 6: **Zero-Shot Relighting.** Our relighting model, trained only on the multi-illumination dataset, generalizes well to out-of-distribution images, as shown on the IIW dataset (first row) and StyleGAN images (second row). It accurately infers scene geometry and lighting. Note that it identifies and turns on the bedside lamps in StyleGAN images despite having no training in bedroom images. This demonstrates the model's strong generalization ability and the model clearly "knows" something about light sources.

relight IIW and StyleGAN-generated images. Despite the significant distribution shift in lighting patterns and room setup, our model accurately identifies luminaires and relights images.

### 4.3 Zero-shot albedo evaluation

Constrained scaling allows us to infer albedo without any decoding (and without any albedo data!) by setting $\alpha = 0$ during inference. We benchmark these albedo estimates using the WHDR metric on the IIW [5] dataset (Section 2). We use WHDR because it is widely used and allows comparisons, but existing literature records significant problems in interpreting the measure [16, 6, 29]. Among other irritating features, the metric seems to prefer odd colors, and can be hacked by heavily quantized albedo maps. As is standard, we obtain lightness by averaging R, G, and B albedo and compute relative lightness of two pixel locations $i_1, i_2$ by comparing to a confidence threshold $\delta$:

$$\widetilde{J}_{i,\delta}(\bar{R}) = \left\{ \begin{array}{ll} 1 & \text{if } \bar{R}_{i_1}/\bar{R}_{i_2} > 1 + \delta \\ 2 & \text{if } \bar{R}_{i_2}/\bar{R}_{i_1} > 1 + \delta \\ E & \text{otherwise} \end{array} \right\} \tag{11}$$

| Methods | labels | Flat | Tune $\delta$ | WHDR |
|---|---|---|---|---|
| Intrinsic Diffusion [29] | CG | No | No | 22.61 |
| Intrinsic Diffusion[29] | CG | Yes | Yes | 17.10 |
| Inverser Render[52] | No | No | No | 21.40 |
| BBA[16] | No | No | Yes | 17.04 |
| Ours | No | No | No | 28.97 |
| Ours | No | No | Yes | 19.09 |
| Ours | No | Yes | Yes | **15.81** |

Table 3: We benchmark our albedo esimation on test set of IIW dataset [5] and compare with others, though the reliability has been questioned by recent papers [16]. Flat denotes postprocessing images with flattening [10]. Despite our model never being trained on albedo maps or CG data, our best configuration significantly outperforms all other methods suggesting our model learns high-quality intrinsic representations

| $\alpha$ | WHDR | | | |
|---|---|---|---|---|
| | $\delta = 0.1$ | | optimal $\delta$ | |
| | w/ F | w/o F | w/ F | w/o F |
| 1e-2 | **17.64** | **28.97** | **15.81** | **19.09** |
| 5e-3 | 18.93 | 31.81 | 16.02 | 19.53 |
| 1e-3 | 18.00 | 29.77 | 15.84 | 19.13 |
| 5e-4 | 18.04 | 29.62 | 15.85 | 19.12 |

Table 4: We conduct ablation experiments to assess the impact of $\alpha$ on the quality of albedo. "w/F" and "w/o F" denote post-processing images with and without flattening [10], respectively. The setting of $\delta = 0.1$ and w/o F is the most affected by $\alpha$. Despite this, all values of $\alpha$ achieve high performance in our optimal configurations.

Figure 7: Qualitative Comparison of **Emergent Albedo from Latent Intrinsics** on the IIW Dataset. Although our model has never been trained on any albedo-like maps, it effectively removes the effects of external light and dark shadows from the input. In contrast, Intrinsic Diffusion [29], a supervised method trained on large computer graphics data, often produces color-drifted estimations, likely due to the domain shift between CG data and real images. Observe the subdued lighting around the mirrors (top row, right) in our recovered albedo. Also, pay attention to all the details inside the refrigerator, which are visible in our recovered albedos (bottom row; right) compared to intrinsic diffusion. For comparison, we also display naive flattening (in the second column), which by itself cannot effectively reduce the strong lighting effects.

The resulting classification (one lighter than two; two lighter than one; equivalent) is then compared to human annotations $J$ using the confidence score $w_i$ for each annotation pair. We report WHDR on the IIW test split in Table 3 to facilitate comparison with other approaches. Since our model is not trained with any albedo maps or computer-generated images, we need to adjust the threshold for the optimal performance. Following prior work, we optimize $\delta$ on the training split, which significantly improves our performance from 28.97 to 19.09. Additionally, we enhance our performance by post-processing our albedo map using flattening [10], an optimization technique to further reduce color variations. With this improvement, our results reach 15.81, substantially outperforming the intrinsic diffusion model [29], a diffusion-based albedo regression model trained on computer graphics data. In Figure 7, we show some qualitative comparisons to intrinsic diffusion. We observe that our method effectively removes external lighting effects and does not suffer from color drift due to domain gap unlike intrinsic diffusion, which is trained on CG data.

**Sensitivity to light changes:** Albedo are scene properties that are independent of lighting changes. In Figure. 8, we qualitatively assess this characteristic by varying lighting conditions, comparing our approach with the state-of-the-art supervised method, Intrinsic Diffusion [29]. Our method demonstrates consistent and accurate estimations that remain stable even under extreme lighting variations. In contrast, Intrinsic Diffusion [29] shows significant deviation from the natural color distribution and are sensitive to lighting changes.

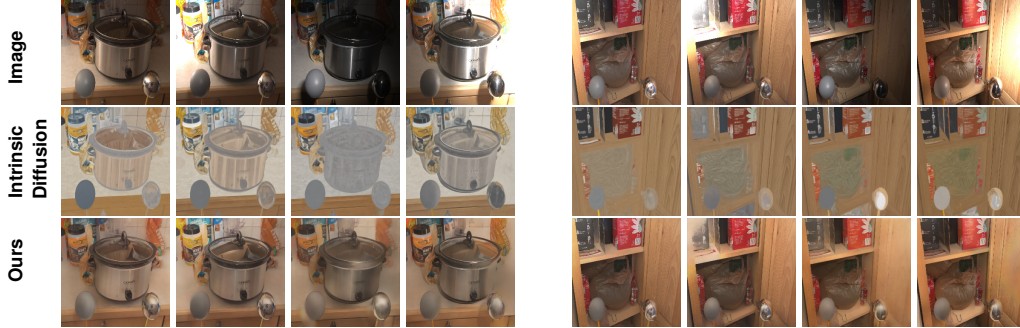

Figure 8: Qualitative comparison of albedo stability under varying lighting conditions. Images shown are from the multi-illumination dataset test split. The top row features images under different lighting environments. The middle row presents estimated albedos obtained from Intrinsic Diffusion [29], while the bottom row shows the recovered albedos from the latent intrinsic representation. Intrinsic Diffusion has large color drift and is sensitive to changes in lighting. In contrast, the **albedos recovered from latent intrinsics remain stable under lighting changes, even in extreme conditions.**

## 5  Discussion, Limitations and Future Work

Our method presents an important advancement in image relighting by demonstrating that intrinsic properties such as albedo can emerge naturally from training on relighting tasks without explicit supervision. This finding simplifies the relighting process, eliminating the need for detailed geometric and surface models and enhancing the model's ability to generalize across diverse and unseen scenes. By encoding scene and illumination properties as latent variables, we achieve accurate and flexible relighting. Our findings will have implications for various fields such as virtual reality and cinematic post-production. This approach reduces the learning process's complexity and offers a new perspective on designing deep learning models to capture and utilize intrinsic scene properties. These findings can guide future research toward a more efficient and scalable relighting approach, encouraging the development of models that can handle various lighting conditions and scene complexities.

The current taxonomy of surface intrinsics—typically, depth, normal, albedo, and perhaps specular albedo and roughness—is quite limiting (compare human language for surface properties [4]). Our method, which computes latent intrinsic and extrinsic representations from images and combines these to transfer lighting conditions across scenes, captures physical concepts like luminaire and albedo without explicit physical parametrization. This ability to represent significant image effects without choosing a surface model offers substantial flexibility.

However, our method has several limitations. It relies on pairs of relighted data captured in the same scene, which can be resource-intensive to obtain. Additionally, it does not cope well with saturated pixel values common in LDR images. The intrinsic information being latent is another limitation since many applications require explicit intrinsic information like depth and normals.

Nonetheless, there is good evidence that explicit intrinsic information can be extracted from our latent intrinsics. Our method clearly "knows" albedo, and this information can be elicited without examples. Similarly, it "knows" something about luminaires, such as their locations and effects. It is intriguing to speculate that it "knows" other information relevant to relighting, such as depth or surface microstructure. Future work will pursue this line of inquiry and also focus on developing a purely unsupervised framework to infer intrinsic and extrinsic properties from collections of in-the-wild images. This will include refining probing techniques for better extraction of explicit intrinsics and identifying additional intrinsic properties crucial for relighting that do not align with the current taxonomy. We believe this will improve the applicability and robustness of our approach, making it suitable for a wider range of real-world scenarios.

## Acknowledgment

AB thanks Stephan R. Richter for the discussions that led to the consideration of intrinsic images as latent variables. This material in part is based upon work supported by the National Science Foundation under Grant No. 2106825, and by a gift from Boeing.

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

# A Experiment Details

**Training Details** We train our model with a batch size of 256 for 1,000 epochs using the AdamW optimizer, with a constant learning rate of 2e-4 and a weight decay ratio of 1e-2. To improve the semantic representation, we corrupt images with Gaussian noise during the first 400 epochs and follow Karras et al. [25] to sample the standard deviation $\sigma$ with $\ln(\sigma) \sim \mathcal{N}(-1.2, 1.2^2)$. In the later 600 epochs, we turn off the Gaussian noise to focus on enhancing the image quality. We train our model with 4A40 and a complete training requires 40 hours.

**Model Details** Our autoencoder employs a U-Net architecture, incorporating residual convolutional blocks as the fundamental components. Each block is composed of two convolutional layers, group normalization, and a nonlinear activation function. The structure specifies [1, 2, 2, 4, 4, 4] blocks at each resolution level, starting from a resolution of 256, with the resolution halving after each level. The corresponding configurations for latent channels at these levels are [32, 64, 128, 128, 256, 512].

The intrinsic features, denoted as $\boldsymbol{S}_{s,i}^l$, are gathered from the output of the final block at each resolution level, starting from a resolution of 128x128 down to the bottleneck. For generating extrinsic features $\boldsymbol{L}_s^l$, multiple MLP layers are applied to the bottleneck features of the encoder, followed by averaging across all spatial features. We limit the channel number of the extrinsic features to 16 to prevent them from conveying high-frequency components.

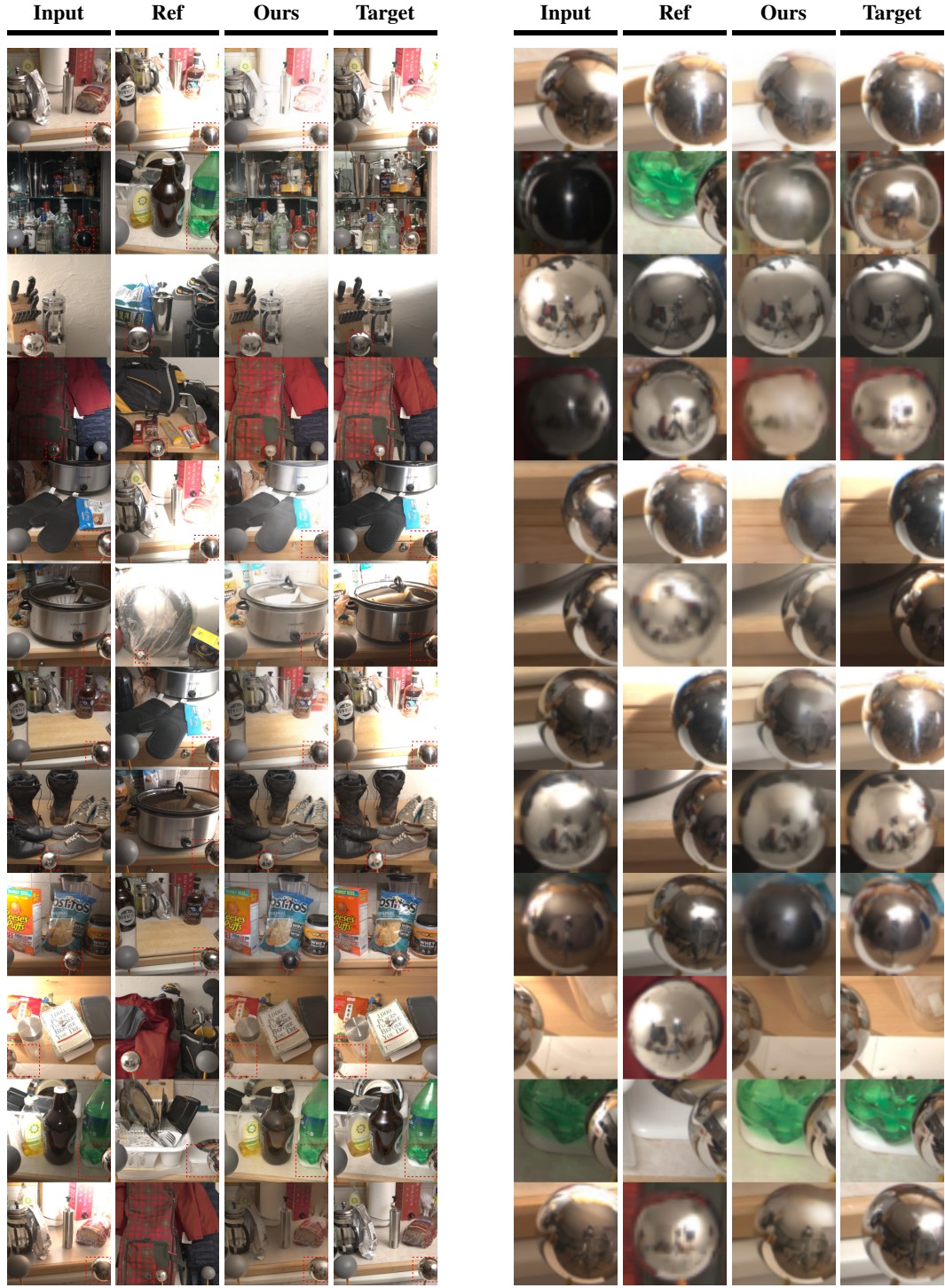

Figure 9: We visualize more examples for the image relighting task in multi-illumination dataset[38]. **Right**: Zoomed-in view of the chrome ball used as a probe to evaluate detail preservation in the environment map.

