# OpenReview forum: "Latent Intrinsics Emerge from Training to Relight"
_NeurIPS.cc/2024/Conference — NeurIPS 2024 spotlight_

### Official Review · Reviewer_n7pv · 2024-07-12

**Soundness:** 4
**Presentation:** 4
**Contribution:** 4
**Rating:** 7
**Confidence:** 4

**Summary:**

This paper proposes a fully data-driven relighting method applicable to images and real scenes. The approach requires only paired images of the same scene under different illumination as inputs. The trained model can also produce albedo-like maps even though it is not trained without such supervision. The experimental results show that it significantly outperforms unsupervised baselines while being competitive with baselines that use supervision.

**Strengths:**

1. The idea and the overall achieved goal are quite novel. Training models that can do relighting without any supervision on ground-truth intrinsic properties (albedo, depth, normal, lighting, etc.) is quite challenging and useful.

2. The presentation is clear. The overall methodology is quite straightforward, with all the details (inputs, outputs, intermediate steps, losses) well-explained and self-contained.

3. The experiments are sufficient.

**Weaknesses:**

1. The paper proposes several novel regularizations, e.g. Eq 6 and Eq 9, which are not ablated. It would be good to see how much these losses improve the model's accuracy.

**Questions:**

Please see the weaknesses.

**Limitations:**

The limitations and potential negative societal impact are discussed properly.

---

> ### Author Rebuttal · Authors · 2024-08-07
>
> We appreciate Reviewer n7pv's recognition of the novelty and usefulness of our work, particularly in training models for relighting without supervision on ground-truth intrinsic properties. We are also grateful for the positive feedback on the clarity and straightforwardness of our presentation, and the comprehensiveness of the methodology and experiments.
>
> **Regarding Ablation of regularization terms**
>
> Thank you for suggesting additional ablation experiments. In response, we trained a model using our optimal configuration ($\alpha=0.5$) for the relighting experiment, excluding the regularization terms in Eq. 6 and Eq. 9 from the training objectives. The results, presented in the attached tables, clearly show that the proposed regularization terms significantly improve performance in both relighting and albedo estimation tasks.
>
> |Relight Config |Raw Output|Color Correction|
> |---|---|----|
> | | RMSE SSIM| RMSE SSIM|
> |w/ regularization | 0.297 0.473 | 0.222 0.571|
> |w/o regularization | 0.315 0.462 | 0.232 0.550|
> Table 1: Ablation experiments of regularization term for image relighting.
>
> \_
>
> |Albedo Config |$\delta = 0.1$|optimal $\delta$|
> |---|---|----|
> |w/ regularization | 31.81 | 19.53 |
> |w/o regularization | 32.56 | 19.67 |
> Table 2: Ablation experiments of regularization term for unsupervised albedo estimation.

---

> > ### Comment · Reviewer_n7pv · 2024-08-10
> >
> > Thanks for the update! The above reply resolves my concerns and I have no further questions. I will keep my score as accept (7).

---

> > > ### Author Response · Authors · 2024-08-12
> > >
> > > Thank you for acknowledging our response, and we're glad we were able to address your concerns. We will include the results of the ablation studies in our final version.

---

### Official Review · Reviewer_t1Cn · 2024-07-12

**Soundness:** 3
**Presentation:** 3
**Contribution:** 3
**Rating:** 6
**Confidence:** 4

**Summary:**

This paper proposes a fully data driven method to perform scene relighting that does not require any groundtruth lighting supervision. As such, it allows for scene relighting by training only on real paired images of the same scene under different illuminations rather than requiring synthetic data that contains the groundtruth lighting environments, which improves the generalizability and performance on real-world scenes. The method involves learning both intrinsic and extrinsic features from each scene, where the intrinsics represent the albedo and geometry and the extrinsics represent the lighting. The authors design a fairly simple but well thought out pipeline with several loss functions that encourage the intrinsics to be consistent for the same scene under different illumination, self reconstruction losses that reconstruct the original image using its own intrinsic and extrinsic features, and relighting loss functions that involve swapping extrinsic features between image pairs to perform scene relighting. The authors also design a constrained scaling mechanism that prevents too much information outside of the lighting information in the target image from being transferred to the source image. Experiments demonstrate that the relighting performance greatly outperforms other unsupervised methods that don't require groundtruth lighting and is comparable or slightly better depending on the metric than the existing state of the art supervised scene relighting methods. Ablations for the constrained scaling are also provided, which is helpful to understand the importance of this technical contribution.

**Strengths:**

Experiments in the paper are quite thorough and demonstrate state-of-the-art performance for scene relighting, most notably being comparable to methods that require full light supervision and greatly outperforming unsupervised methods. Ablation studies on the constrained scaling are also helpful for understanding this contribution.

The model and pipeline itself is quite simple and straightforward yet effective and easy to follow. The loss functions align with intuition and are carefully designed.

The ability to estimate albedo from a scene is an important feature that has many implications for downstream tasks, and the authors achieve state-of-the-art performance for albedo estimation.

**Weaknesses:**

One experiment that would be interesting to see is the level of albedo consistency. Since the authors design a loss function that encourages intrinsic features of the same scene under different illuminations to be similar, it would be good to see how well that design performs empirically, both quantitatively and qualitatively.

Some citations are missing for the image-based relighting domain, namely:

1. Towards High Fidelity Face Relighting with Realistic Shadows (CVPR 2021)
2. Learning to Relight Portraits for Background Replacement (SIGGRAPH 2021)
3. Face Relighting with Geometrically Consistent Shadows (CVPR 2022)
4. Lumos: Learning to Relight Portrait Images via a Virtual Light Stage and Synthetic-to-Real Adaptation (SIGGRAPH Asia 2022)
5. DiFaReli: Diffusion Face Relighting (ICCV 2023)

**Questions:**

As I mentioned above, the experiments in this paper are quite thorough and I'm mostly interested to see whether the estimated albedo is consistent for the same scene under different illuminations. It would be good to compare this with other methods as well. Otherwise, please add the missing citations for the image-based relighting domain as there are many recent methods in that area.

**Limitations:**

The limitations section is well thought out and touches on many important points. Relighting often does not have significant negative societal impacts since only the illumination is edited, but as with any other editing work there is always the potential to generate fake content. Perhaps this could be mentioned in the potential negative societal impact section.

---

> ### Author Rebuttal · Authors · 2024-08-07
>
> We appreciate Reviewer t1Cn's feedback and are grateful for their recognition of the simplicity and effectiveness of our proposed method, the thoroughness of our experiments, and our approach to unsupervised training. We are pleased to see the acknowledgment of our design choices, the clarity and straightforwardness of our paper's writing, and the practical implications of our method for downstream tasks. We now respond to the reviewer's questions.
>
>
> **Regarding Consistency of Our Albedo Predictions**
>
> We conducted an experiment to evaluate the consistency of albedo predictions under varying lighting conditions. Specifically, we estimated albedo for the same scene under different lighting settings, as provided in the MIT dataset. Qualitative visualizations are included in the attached PDF. For comparison, we also evaluated the state-of-the-art supervised intrinsic estimator, Intrinsic Image Diffusion (IID) [1]. Notably, IID requires 10 independent estimations followed by averaging to produce a single image estimation. Due to the computational expense of this process, we assessed albedo consistency under only five different lighting settings.
>
> Our method, on the other hand, produces consistent results with just a single forward pass. Our findings demonstrate that our approach achieves stable albedo estimation despite not utilizing any albedo-like maps as supervision. Also, while IID often exhibits significant color drift due to its reliance on synthetic training data, our method maintains color fidelity, as shown in our qualitative figures. To quantify the albedo stability under varying lighting conditions, we report the mean deviation and standard deviation on albedo map, normalized per scene,
>
> |Results |Mean Deviataion| Standard Deviation|
> |---|---|---|
> |IID | 0.054| 0.063|
> |Ours | **0.046**| **0.053**|
>
> Our method produces more stable albedo estimations under varying lighting conditions compared to the supervised IID approach.
>
>
> [1] Kocsis et al., Intrinsic Image Diffusion for Indoor Single-view Material Estimation. CVPR 2024.
>
> **On Missed citations**
>
> Thank you for bringing these papers to our attention. We will ensure that these citations are added to the related work section in our final draft.

---

> > ### Comment · Reviewer_t1Cn · 2024-08-09
> >
> > Thank you for your detailed rebuttal! My concerns are addressed and I am convinced by the authors' rebuttal with regard to other reviewers' concerns as well. I will maintain my Weak Accept rating. Please be sure to include missing details in the final version, especially the ablation tables.

---

> > > ### Author Response · Authors · 2024-08-12
> > >
> > > Thank you for acknowledging our response, and we're glad we could address your concerns. We will incorporate your suggested references, along with the additional visualizations and results, in our final version.

---

### Official Review · Reviewer_VUQQ · 2024-07-14

**Soundness:** 3
**Presentation:** 2
**Contribution:** 3
**Rating:** 6
**Confidence:** 4

**Summary:**

The paper proposes a 2D relighting pipeline based on latent space manipulation, which is purely data-driven without explicit intrinsic representations such as geometry and materials. Given a single image input, the proposed model recovers latent variables representing scene intrinsic properties and latent variables representing lighting, enabling applications like 2D relighting based on a single reference image and albedo estimation. The proposed methods are validated on a held-out dataset, demonstrating its generalization capability and precision in real-world scenarios.

**Strengths:**

1. The paper is well-written and easy to understand.
2. The idea of relighting images with latent latent intrinsic and without explicit explicit intrinsic representations is interesting and novel. The paper shows impressive results on the test datasets and outperforms the baselines for relighting and albedo estimation.
3. The proposed method doesn't need explicit lighting supervision and is purely data-driven, which indicates the method may have the potential to be scaled up in the future.

**Weaknesses:**

1. using latent intrinsic to relight means the users lose the ability to control the relighting in a fine-grained way. For example, the user can't precisely control the lighting intensity and directions.

2. The proposed method doesn't show enough generalization ability. In Sec. 4.2, the authors first mention that they train the model on the Multi-illumination training dataset and then test it on the Multi-illumination test dataset. Then, when the authors validate the method on StyLitGAN images, the authors mention that they train the model again on StyLitGAN images. (as said in L213). It seems to mean that the trained model can't be used to test out of the distribution images directly. To show enough generalization ability, the model should be able to test on unseen real images directly after training ends.

**Questions:**

What if the authors use the trained model to test some unseen images on the internet? Will the model still show good relighting ability? Can the checkpoint trained on the Multi-illumination dataset be used to infer StyLitGAN images? Some results would be appreciated.

**Limitations:**

The limitations have been well discussed by the authors in the paper.

---

> ### Author Rebuttal · Authors · 2024-08-07
>
> We appreciate Reviewer VUQQ's feedback and are grateful for their recognition of the novelty of our proposed method, our approach to unsupervised training, and the quality of our results and paper writing. We now address the specific concerns and questions raised by the reviewer.
>
>
> **Concern: Lack of Fine-Grained Control in Relighting because of latent lighting representation**
>
> While it is true that using latent lighting representation for relighting may limit the ability to achieve fine-grained control over specific lighting parameters such as intensity and direction, this limitation is not unique to our approach. Current explicit parametric methods, like spherical harmonics or spherical Gaussians, also face challenges in providing precise control over these aspects, often due to the complexity and computational cost involved.
>
> Our approach, however, offers a significant advantage by simplifying the relighting procedure. By learning a high-dimensional latent representation, we bypass the need for detailed manual tuning of lighting parameters, making the process more efficient and accessible. Additionally, we have demonstrated the ability to interpolate in the latent space to achieve various lighting effects, as shown in Figure 4. This capability suggests that while explicit fine-grained control is limited, our method provides a practical and scalable solution for many real-world applications where such precision may not be critical. Future work will explore ways to enhance control, potentially by mapping latent representations to explicit lighting parameters for users who require finer-detailed adjustments.
>
>
> **Concern: Generalization Ability**
>
> We respectfully disagree with the reviewer's assessment regarding the generalization capabilities of our model. Our paper demonstrates two key aspects of the model's generalization ability, which we detail below:
>
> 1) **Generalization to Unpaired Images:** While our model is trained on paired images captured under the same scene, we evaluate its performance on unpaired images. This is illustrated in the first two columns of Figure 3, where the input and reference images originate from different scenes. We opted for paired training data due to practical scalability considerations, as using unpaired data necessitates the calibration of exact extrinsic lighting conditions across different scenes. Our model significantly outperforms comparable approaches in accurately rendering relit patterns while preserving environmental details.
>
> 2) **Generalization to Out-of-Distribution Images:** We trained our model on the Multi-Illumination dataset and directly tested it on the IIW dataset for albedo estimation, without any additional training or prior exposure to the IIW dataset. Despite the considerable distribution shift between these datasets, our method consistently outperforms other approaches in albedo estimation, even without using any albedo-like supervision.
>
>
> To further substantiate our model's generalization capability, we provide additional results as suggested by the reviewer. We tested the relighting performance of models trained on different datasets, including demonstrating plausible relighting of StyLitGAN and IIW images using models trained on the MIT Multi-Illumination dataset. These experiments show the model's ability to generalize to unseen images using two sources of extrinsics: (a) estimating extrinsics from in-distribution images and (b) estimating extrinsics from out-of-distribution images. Visualizations of these results are included in the attached PDF.
> While these results demonstrate plausible relighting, we acknowledge that they are not perfect, primarily due to the distinct nature of the distributions. The MIT Multi-Illumination dataset does not feature visible luminaires or light sources, whereas StyLitGAN images may include visible luminaires, with the quality dependent on StyleGAN's generative capabilities. We believe that expanding our training dataset will enhance rendering quality, as our approach can easily scale due to its lack of reliance on supervision.
>
> We appreciate the reviewer's feedback and questions, and we will incorporate these clarifications into the final version of our manuscript for improved clarity.

---

> > ### Comment · Reviewer_VUQQ · 2024-08-09
> >
> > Thanks for the authors' response.  My concerns have been addressed, and I will raise my score to 6.
> >
> > I also have some further suggestions for the authors, which are purely recommendations and do not affect the evaluation of this paper:
> >
> > 1. I suggest that the authors consider building a project website in the future. This would allow them to showcase more results and, more importantly, present video results of Latent Extrinsic Interpolation (as shown in Fig. 4), which could significantly enhance the paper's presentation. (BTW, can the current methods get consistent video results?)
> >
> > 2. Regarding the "Generalization to Out-of-Distribution" problem, as the authors have acknowledged, there is a noticeable performance drop when dealing with data that has different distributions. In addition to expanding the dataset, as the authors mentioned, combining the current model designs with large generative models, such as Stable Diffusion, might improve performance and generalization. This could be an interesting direction for future work.

---

> > > ### Author Response · Authors · 2024-08-12
> > >
> > > Thank you for acknowledging our response, and we’re pleased that we were able to address your concerns. We also appreciate your suggestion of creating a project webpage for an interactive demonstration. Your follow-up idea of leveraging the richer structural priors of the generative model to enhance performance sounds both interesting and promising—thank you for proposing it.

---

### Author Rebuttal · Authors · 2024-08-07

We thank the reviewers for their positive and constructive feedback on our paper. Reviewers consistently highlighted the "novelty" [VUQQ, n7pv] of our approach, particularly appreciating the method's ability to perform relighting using latent intrinsic properties without explicit representations, described as "interesting and novel" [VUQQ]. The paper was noted for being "well-written and easy to understand" [VUQQ], with a "clear" [n7pv] and "straightforward" [n7pv] methodology. Reviewers also commended the "thorough" [t1Cn] and "sufficient" [n7pv] experiments, which demonstrate "state-of-the-art performance" [t1Cn] in scene relighting and albedo estimation and "outperforms baselines" [VUQQ, t1Cn]. We are extremely delighted to receive such feedback. Thank you!

Further positive points include the "simplicity and effectiveness" [t1Cn] of our model and pipeline, the careful design of loss functions that "align with intuition" [t1Cn], and the important feature of estimating albedo, which has significant "implications for downstream tasks  [t1Cn]." The method's "data-driven" nature, which "doesn't need explicit lighting supervision," suggests potential for future scalability [VUQQ]. Overall, the reviewers appreciated the challenging and useful nature of training models for relighting without supervision on ground-truth intrinsic properties [n7pv].

With the clarifications provided (detailed responses to each reviewer below), we hope our paper stands as a meaningful contribution to the community. Here is a summary of our responses to the reviewers' questions and concerns, along with summaries of results and figures included in our rebuttal PDF:

- For VUQQ: We provide visualizations of relighting unseen images to demonstrate the generalization and robustness of our approach.

- For t1Cn: We offer results on albedo prediction and its variance under various lighting conditions to demonstrate the stability of the learned intrinsic representation.

- For n7pv: We present an ablation study excluding regularization losses, demonstrating their necessity for improved relighting and albedo prediction.

---

### Decision · Program_Chairs · 2024-09-25

**Decision:**

Accept (spotlight)

**Comment:**

This paper initially received unanimous positive scores. Reviewers appreciated the clear presentation, the thoroughness of the evaluation, and they praised the approach to relighting using the emergent scene intrinsics. Reviewers also raised concerns related to, e.g., generalization to different datasets, consistency of albedo predictions, and the effect of regularization terms. The authors provided a rebuttal, and after the reviewer discussion all reviewers agreed that the paper should be accepted. The authors should include the clarifications from the rebuttal in the paper.